# Clinical and Molecular Characterization of Nine Novel Antithrombin Mutations

**DOI:** 10.3390/ijms25052893

**Published:** 2024-03-01

**Authors:** Judit Kállai, Réka Gindele, Krisztina Pénzes-Daku, Gábor Balogh, Réka Bogáti, Bálint Bécsi, Éva Katona, Zsolt Oláh, Péter Ilonczai, Zoltán Boda, Ágnes Róna-Tas, László Nemes, Imelda Marton, Zsuzsanna Bereczky

**Affiliations:** 1Division of Clinical Laboratory Science, Department of Laboratory Medicine, Faculty of Medicine, University of Debrecen, 4032 Debrecen, Hungary; kallai.judit@med.unideb.hu (J.K.); gindele.reka@med.unideb.hu (R.G.); kpenzes@med.unideb.hu (K.P.-D.); balogh.gabor@med.unideb.hu (G.B.); bogati.reka@med.unideb.hu (R.B.); ekatona@med.unideb.hu (É.K.); 2HUN-REN-UD Cell Biology and Signaling Research Group, Faculty of Medicine, University of Debrecen, 4032 Debrecen, Hungary; 3Department of Medical Chemistry, Faculty of Medicine, University of Debrecen, 4032 Debrecen, Hungary; bbalint@med.unideb.hu; 4Department of Anesthesiology and Intensive Care, Faculty of Medicine, University of Debrecen, 4032 Debrecen, Hungary; zsolah@med.unideb.hu; 5Jósa András Teaching Hospital of University of Debrecen, 4400 Nyíregyháza, Hungary; ilonczai@med.unideb.hu; 6Department of Internal Medicine, Faculty of Medicine, University of Debrecen, 4032 Debrecen, Hungary; zboda@med.unideb.hu; 7Department Medical Center of the Hungarian Defence Forces, National Hemophilia Center and Hemostasis, 1134 Budapest, Hungary; ronat@t-online.hu (Á.R.-T.); lnemes@t-online.hu (L.N.); 8Department of Medicine, Albert Szent-Györgyi Medical School, University of Szeged, 6720 Szeged, Hungary; marton.imelda@med.u-szeged.hu; 9Department of Transfusion Medicine, Albert Szent-Györgyi Medical School, University of Szeged, 6720 Szeged, Hungary

**Keywords:** antithrombin, antithrombin deficiency, *SERPINC1* mutation, expression study, surface plasmon resonance, in silico methods

## Abstract

Antithrombin (AT) is the major plasma inhibitor of thrombin (FIIa) and activated factor X (FXa), and antithrombin deficiency (ATD) is one of the most severe thrombophilic disorders. In this study, we identified nine novel AT mutations and investigated their genotype–phenotype correlations. Clinical and laboratory data from patients were collected, and the nine mutant AT proteins (p.Arg14Lys, p.Cys32Tyr, p.Arg78Gly, p.Met121Arg, p.Leu245Pro, p.Leu270Argfs*14, p.Asn450Ile, p.Gly456delins_Ala_Thr and p.Pro461Thr) were expressed in HEK293 cells; then, Western blotting, N-Glycosidase F digestion, and ELISA were used to detect wild-type and mutant AT. RT-qPCR was performed to determine the expression of AT mRNA from the transfected cells. Functional studies (AT activity in the presence and in the absence of heparin and heparin-binding studies with the surface plasmon resonance method) were carried out. Mutations were also investigated by in silico methods. Type I ATD caused by altered protein synthesis (p.Cys32Tyr, p.Leu270Argfs*14, p.Asn450Ile) or secretion disorder (p.Met121Arg, p.Leu245Pro, p.Gly456delins_Ala_Thr) was proved in six mutants, while type II heparin-binding-site ATD (p.Arg78Gly) and pleiotropic-effect ATD (p.Pro461Thr) were suggested in two mutants. Finally, the pathogenic role of p.Arg14Lys was equivocal. We provided evidence to understand the pathogenic nature of novel *SERPINC1* mutations through in vitro expression studies.

## 1. Introduction

The antithrombin gene (*SERPINC1*) is mapped to chromosome 1q23.1–23.9 and comprises seven exons and six introns spanning a total of 13.5 kb of genomic DNA [1]. The signal peptide of 32 amino acids directs the translocation of the protein to the endoplasmic reticulum, where it is folded and subjected to two post-translational modifications: formation of 3 intramolecular disulfide bonds and N-glycosylation at Asn128, 167, 187 and 224 [2,3,4]. Mature antithrombin (AT) is a 58 kDa plasma glycoprotein that consists of 432 amino acid residues with three intramolecular disulfide bridges at Cys40-160, Cys53-127 and Cys279-462. Antithrombin belongs to the superfamily of serine protease inhibitors (SERPINs). It is the major plasma inhibitor of thrombin and activated factor X (FXa); however, it also inactivates FIXa, FXIa, FXIIa and FVIIa. In the presence of heparin or heparan sulfate proteoglycans, the rate of inhibition is accelerated.

Antithrombin deficiency (ATD) is the most severe inherited thrombophilia according to our present knowledge, and it may be inherited or acquired. A decrease in functional antithrombin in plasma results in an increased risk of thromboembolism.

Two major forms of ATD are recognized: type I and II. Type I (quantitative) deficiency, which has been identified only in the heterozygous state, is characterized by a reduction in AT concentration to ~50% of normal. In contrast, type II (qualitative) deficiencies are characterized by the presence of a dysfunctional protein in the plasma [5]. The defect may affect the reactive site (II RS) or the heparin binding site (II HBS), or it can cause a pleiotropic effect (II PE) [6,7]. The prevalence of ATD has been estimated as approximately 1 per 2000 individuals. However, in patients who develop thrombosis, the prevalence is increased to between 1 in 20 and 1 in 200 individuals [8,9]. According to the Human Gene Mutation Database (http://www.hgmd.cf.ac.uk, accessed on 11 November 2023), *n* = 529 different mutations have been reported thus far in *SERPINC1*. In most cases, missense/nonsense mutations have been registered (*n* > 231), but splicing variants, small deletions and insertions have also been presented. Detailed clinical and molecular characterizations of the identified mutations provide information to understand the pathophysiology of the disease and help in the classification of novel mutations according to pathogenicity. Extensive in vitro characterization of AT mutations has been demonstrated in only a few studies [10,11,12,13,14]. We identified nine novel AT gene mutations in patients with thrombosis. In this study, we investigated the genotype–phenotype correlations and the molecular characteristics of these mutations.

## 2. Results

### 2.1. Clinical and Laboratory Characteristics of Antithrombin-Deficient Patients

Patients with ATD were diagnosed at our tertiary center for hemostasis diseases between 2016 and 2021. Altogether, *n* = 311 patients were genetically investigated for ATD during this period. The majority of them carried the founder AT Budapest3 (p.Leu131Phe) mutation. In our cohort, however, we have found nine mutations within *SERPINC1* that have not been reported before. These mutations are missing from the different databases (HGMD Professional and ClinVar, https://www.ncbi.nlm.nih.gov/clinvar, accessed on 11 November 2023), or they have been reported by us [15]. We filtered our results against data deposited in the 1000 Genomes Project (http://www.1000genomes.org, accessed on 11 November 2023), and they were not considered polymorphisms. All patients were heterozygous for these novel mutations (Table 1). Two patients (father and son) carried the c.41G > A (p.Arg14Lys) mutation. In their plasma samples, decreased heparin cofactor anti-FXa AT activity (hc-antiFXa) and decreased progressive anti-FXa AT activity (p-antiFXa) were measured. The AT antigen concentration was also decreased, suggesting a rather quantitative ATD. The proband suffered a DVT at the age of 16 years, and no recurrence was registered. His father, a carrier of the same variant, has not reported a thrombotic episode as yet. The index patient was recommended to receive long-term VKA treatment; his father is not anticoagulated.

The second mutation (c.95G > A, p.Cys32Tyr) was carried by one female patient, who suffered her first thrombotic episode, a DVT with PE, at the age of 20 years during pregnancy. We registered two recurrences: a mesenterial vein thrombosis occurring at the age of 41 years and a PE several years later. The patient was on VKA after the first and second thrombosis for one year each; however, after the third thrombotic episode, she was put on lifelong anticoagulation, starting with VKA and then switching to rivaroxaban (20 mg OD). The patient’s family history is positive for thrombosis: her three brothers also suffered from DVT, and two of them had a fatal PE. However, they were not investigated for thrombophilia at that time. The patient’s daughter suffered from DVT after delivery. The hc-antiFXa and p-antiFXa AT activity values of the patient were low, and a proportionally decreased AT antigen level was measured.

The third mutation (c.232C > G, p.Arg78Gly) affects the heparin-binding region of AT. The carrier was a young female patient with endometriosis. She was examined due to infertility, but she had never had a thrombotic episode. Her family history was also negative for thrombosis. Before an in vitro fertilization (IVF) procedure, she was diagnosed with ATD. During the IVF procedure and her pregnancy, she was treated by LMWH, but no long-term anticoagulation was indicated. The hc-antiFXa AT activity of the patient was decreased, while normal p-antiFXa AT activity and a normal AT antigen level were detected suggesting type II HBS AT deficiency.

The fourth mutation (c.362T > G, p.Met121Arg) was carried by a young female who had had one episode of DVT at the age of 34 years. Hc-antiFXa and p-antiFXa AT activity and AT antigen values were proportionally low, suggesting type I ATD. The patient was put on long-term VKA anticoagulation.

The fifth mutation (c.734T > C, p.Leu245Pro) was carried by a male patient with proximal DVT at the age of 42 years. He had no provoking factors in his case history. Hc-antiFXa and p-antiFXa AT activity and AT antigen values were proportionally low, suggesting type I ATD. Upon diagnosis, he was put on warfarin for long-term anticoagulation. No recurrent thrombotic events were registered.

The sixth mutation (c.809delT, p.Leu270Argfs*14) was detected in a male patient who had had 5 episodes of DVT. His first DVT was diagnosed at the age of 16 years. From the clinical point of view, this case showed the most severe thrombotic phenotype. Hc-antiFXa and p-antiFXa AT activity and AT antigen values were proportionally low, suggesting type I ATD. The patient is on lifelong anticoagulation and is currently taking rivaroxaban (20 mg OD).

The seventh mutation (c.1349A > T, p.Asn450Ile) was carried by a family with four members, of whom two had already suffered from DVT at the ages of 23 and 40 years. Their hc-antiFXa and p-antiFXa AT activity and AT antigen values were proportionally low, suggesting type I ATD. Patient 7_1 in Table 1 has not suffered from thrombosis as yet; however, she has had four pregnancies, among which the first ended with a spontaneous abortion. During the three subsequent pregnancies, the patient was treated with LMWH and AT concentrate, and those pregnancies were successful. No long-term anticoagulation was introduced for her. The sister of the index patient (7_2) was also treated with LMWH and AT concentrate during pregnancy; however, due to DVT in her case history, she was put on long-term VKA anticoagulation. Patient 7_3 is also anticoagulated with VKA, while patient 7_4 is not, and he has not suffered from thrombosis as yet.

The eighth mutation (c.1367-1368delGCinsCTACA, p.Gly456delins_Ala-Thr) was detected in a female patient who had had a pregnancy-associated DVT at the age of 27 years, which was followed by a second (unprovoked) episode 5 years later. Hc-antiFXa and p-antiFXa AT activity and AT antigen values were proportionally low, in accordance with a type I ATD. The patient is on lifelong anticoagulation with apixaban (5 mg BID).

Finally, the ninth mutation (c.1381C > A, p.Pro461Thr) was detected in three unrelated patients. One of them had suffered an unprovoked thrombosis followed by pulmonary embolism at a relatively old age (62 years), and he is now on VKA treatment; the other two patients were much younger at the time of their thrombotic events, but both DVTs were provoked. Both patients are on DOACs (apixaban 5 mg BID and rivaroxaban 20 mg OD). Hc-antiFXa and p-antiFXa AT activity values were disproportionally low compared to AT antigen values, suggesting a rather functional AT deficiency.

Concerning laboratory investigations, the basic tests of coagulation were normal for all patients, or at least appropriate for their anticoagulant treatment. Thrombophilia testing showed activated protein C resistance, FV Leiden heterozygosity in two patients (patients No 2_1 and 8_1 in Table 1) and no other detected thrombophilia.

### 2.2. AT Sequence Homology Study with UniProt Database

As indirect evidence of pathogenicity, it is known that disease-causing mutations are found in the strongly conserved positions of proteins [16].

Therefore, we examined AT sequence homology in seven different species at the positions of the nine mutations using the UniProt Database (Figure 1). The investigated amino acid positions are identical in the seven different species (except at position 270, where there is a leucine instead of a proline in three different species; however, they are similar non-polar amino acids with hydrophobic side chains), meaning strongly conserved positions. In the case of *P. troglodytes*, we noted that all positions and their surrounding regions are identical. Position 78 and 121 and their surrounding regions are identical in all seven species. The signal peptide sequence is heterogeneous in the seven species, but it has three conserved parts: a hydrophilic part in the N terminal, a hydrophobic core and a polar C-terminal region. Position 32 is in the conserved polar C-terminal region containing the signal peptide cleavage site.

### 2.3. In Silico Prediction of the Consequences of Missense Mutations

We analyzed the seven AT missense mutations using six different tools available as web servers that can predict their consequences (Table 2). The mutations p.Met121Arg, p.Leu245Pro, p.Asn450Ile and p.Pro461Thr were predicted by all six methods to be pathogenic or disease-causing. The results for the p.Arg78Gly mutant, affecting an amino acid close to the heparin binding site, were inconclusive. It was classified as benign by PolyPhen 2 (both scores) and PhD-SNP but as pathogenic by all other tools. Two missense mutations, p.Arg14Lys and p.Cys32Tyr, are located in the signal peptide sequence of AT. PolyPhen2 and SIFT predicted these mutants to be deleterious, while MutPred2 and MutationTaster designated them as non-pathogenic. PhD-SNP classified the first mutant as benign and the second as disease-causing. We performed an in silico analysis of signal peptide cleavage using SignalP 6.0. for the two mutations located in this sequence. SignalP 6.0 predicted a signal peptide cleavage site between amino acids 32 and 33 for the wild-type protein and the same situation for the p.Arg14Lys mutant, and the prediction indicated high confidence (>0.999). For p.Cys32Tyr, the program still detected a cleavage site after position 32, but the confidence of the prediction was rather low; it was only slightly above 0.5 (0.542).

### 2.4. Detection of Wild-Type and Mutant Antithrombins in the Cell Lysates and in the Media of Transfected Cells

WT AT appeared as a clear band in the cell lysates and in the conditioned media of HEK293 cells at 58 kDa. The positive control band represented AT from the pooled plasma of 5 healthy individuals and appeared as a single band. As expected, no signal from mock transfection could be detected. In the media of transiently transfected HEK293 cells, we detected a clear band at 58 kDa in the cases of p.Arg14Lys, p.Arg78Gly and p.Pro461Thr. However, only a faint band, if any, could be visualized for p.Cys32Tyr, p.Met121Arg, p.Leu245Pro, p.Leu270Argfs*14, p.Asn450Ile and p.Gly456delins_Ala_Thr. In the cell lysates of transiently transfected cells, AT could be visualized in the cases of p.Arg14Lys, p.Arg78Gly, p.Met121Arg, p.Leu245Pro, p.Gly456delins_Ala_Thr and p.Pro461Thr. In the cases of p.Cys32Tyr, p.Leu270Argfs*14 and p.Asn450Ile, AT was practically absent or showed a very faint band (Figure 2A).

Based on the Western blot analysis, we classified the mutants into three groups. The first group included the cases of p.Cys32Tyr, p.Leu270Argfs*14 and p.Asn450Ile, where we detected low AT protein expression both in the conditioned media and in the cell lysates, suggesting a low level of protein synthesis or even a lack of mRNA. In the case of p.Asn450Ile, Western blotting indicated a reduced (approximately 56 kDa) molecular weight for the AT protein. The second group (p.Met121Arg, p.Leu245Pro and p.Gly456delins_Ala_Thr) showed high AT protein expression in the cell lysates but low AT protein expression in the media, suggesting a secretion disorder. The third group (p.Arg14Lys, p.Arg78Gly, p.Pro461Thr) included those AT proteins appearing as strong bands both in the cell lysates and in the media. These mutants are suggested to be functional variants, or they may have a minor effect on AT structure and function.

In the cases of six mutations (p.Arg14Lys, p.Cys32Tyr, p.Arg78Gly, p.Met121Arg, p.Leu245Pro and p.Pro461Thr), stable transfection was also performed in order to gain higher protein concentrations for further investigations (see later). To check the quality of stable transfection, we detected the WT and the six mutant AT proteins by Western blotting in the conditioned media (Figure 2B). The appearance of the bands corresponding to the different mutants was identical to that seen in the Western blot of the transient transfection.

### 2.5. Real-Time Quantitative PCR Analysis

RT-qPCR was performed to determine the expression of AT mRNA in the transiently transfected HEK293 cells in three independent transfections. To detect the efficiency of transfection, we measured β-GAL mRNA expression. The relative quantification of target RNA was achieved by the comparative threshold cycle (CT) method, and the target CT numbers were normalized to OAZ, a housekeeping gene. The mRNA contents of the variants were expressed as values of the ratio relative to WT, and they were as follows: p.Arg14Lys 1.09 ± 0.03; p.Cys32Tyr 1.03 ± 0.04; p.Arg78Gly 1.03 ± 0.04; p.Met121Arg 1.00 ± 0.03; p.Leu245Pro 1.01 ± 0.01; p.Leu270Argfs*14 0.99 ± 0.07; p.Asn450Ile 0.94 ± 0.04; p.Gly456delins_Ala_Thr 0.96 ± 0.05 and p.Pro461Thr 0.98 ± 0.03, respectively. (Data represent the relative quantification (RQ) ± SEM of each transcript normalized to OAZ.) The AT mRNA expression levels of the nine mutants were not significantly different from that of AT-WT according to RT-qPCR. These results suggest that in the case of decreased AT, even in the cell lysates, the mutations do not affect the mRNA level. In the case of p.Cys32Tyr, p.Leu270Argfs*14 and p.Asn450Ile mutations, for which little or no AT was detected in the cell lysates, a defect in protein synthesis is therefore more likely.

### 2.6. AT Antigen and Activity of Mutant AT Proteins Expressed in HEK293 Cells

WT and mutant AT antigen levels were determined by ELISA in duplicates from each of the different transfection reactions. Figure 3A demonstrates the AT antigen concentrations of the mutants in percentages compared to the WT AT, which was considered 100%. AT antigen values were normalized to the transfection efficiency by performing β-GAL measurements.

The AT antigen concentrations in the conditioned media were as follows (number of different transfection reactions in brackets): p.Arg14Lys 131.00 ± 16.39% (*n* = 4); p.Cys32Tyr 26.25 ± 5.72% (*n* = 4); p.Arg78Gly 141.75 ± 29.77% (*n* = 4); p.Met121Arg 210.00 ± 92.11% (*n* = 4); p.Leu245Pro 100.75 ± 45.55% (*n* = 4); p.Leu270Argfs*14 28.33 ± 9.56% (*n* = 3); p.Asn450Ile 27.00 ± 8.50% (*n* = 3); p.Gly456delins_Ala_Thr 43.66 ± 17.42% (*n* = 3) and p.Pro461Thr 147.00 ± 30.09% (*n* = 4), respectively.

The AT antigen concentrations in the cell lysates were as follows (number of different transfection reactions in brackets), p.Arg14Lys 188.25 ± 11.69% (*n* = 4); p.Cys32Tyr 31.33 ± 0.88% (*n* = 3); p.Arg78Gly 134.25 ± 13.14% (*n* = 4); p.Met121Arg 1706.25 ± 585.45% (*n* = 4); p.Leu245Pro 1555.75 ± 568.72% (*n* = 4); p.Leu270Argfs*14 0.00 ± 0.00% (*n* = 3); p.Asn450Ile 20.00 ± 6.55% (*n* = 3); p.Gly456delins_Ala_Thr 64.66 ± 18.67% (*n* = 3) and p.Pro461Thr 262.50 ± 67.22% (*n* = 4), respectively.

As can be seen, AT antigen concentration was low in the cases of p.Cys32Tyr, p.Leu270Argfs*14 and p.Asn450Ile mutants both in the cell lysates and in the media, corresponding to the results obtained by Western blotting. A considerable amount of AT was detected in the cell lysates of the other six mutants. Among them, the AT antigen was very high in the cases of p.Met121Arg and p.Leu245Pro. AT antigen concentrations in the media of the p.Arg14Lys, p.Arg78Gly and p.Pro461Thr groups were similar to that of WT AT; in the case of p.Gly456delins_Ala_Thr, it was low; and in case of p.Met121Arg and p.Leu245Pro, AT antigen concentrations were disproportionally low in the media compared to the AT level in the corresponding cell lysates.

Except for those mutants where the AT antigen concentration was low in the media, heparin cofactor activity and progressive activity from the conditioned media of the transfected cells were determined by amidolytic assay from at least two independent transfections (Figure 3B). The heparin cofactor activity and progressive activity results of the mutants are represented as percentages of WT AT activity, which was considered 100%. The hc-antiFXa values were as follows: p.Arg14Lys 85.16 ± 18.99%; p.Arg78Gly 61.50 ± 8.50%; p.Met121Arg 106.5 ± 26.67%; p.Leu245Pro 110.00 ± 25.07%; and p.Pro461Thr 93.75 ± 44.67%. The p-antiFXa values were as follows: p.Arg14Lys 99.56 ± 30.07%; p.Arg78Gly 67.54 ± 59.85%; p.Met121Arg 133.46 ± 46.10%; p.Leu245Pro 120.14 ± 43.47% and p.Pro461Thr 160.27 ± 67.06%. In the cases of the p.Arg14Lys, p.Met121Arg and p.Leu245Pro mutants AT activity did not show a decrease in the heparin cofactor and progressive assays, and the two assays gave proportional results. In the case of the p.Arg78Gly mutant, AT heparin cofactor activity was only 60% of the wild-type value, and progressive AT activity values showed a large variance, having reduced and normal values as well in the different experiments. In case of p.Pro461Thr, a large difference between hc-antiFXa and p-antiFXa values was demonstrated.

### 2.7. Investigation of p.Arg78Gly and p.Pro461Thr Mutant AT Heparin Binding by Surface Plasmon Resonance

Based on the functional assays, altered heparin binding was hypothesized in the cases of p.Arg78Gly and p.Pro461Thr AT. Therefore, we investigated the heparin-binding characteristics of these two mutants and used WT AT as a control in a purified system using recombinant proteins. The kinetic and affinity parameters of the interaction were calculated based on the sensorgrams obtained at different AT concentrations and averaged for each AT (Figure 4). As expected, the strongest AT–heparin binding was observed in the case of WT protein (K_D_ = 8.49 × 10^−7^ M). The association rate constant (k_a_) was k_a_ = 2.66 × 10^6^ 1/Ms. These data suggest that the formation of the AT–heparin complex occurs most rapidly in the case of WT AT. In the case of the p.Arg78Gly protein, the K_D_ value was 1.69 × 10^−6^ M, and the association rate constant (k_a_) was k_a_ = 1.15 × 10^3^ 1/Ms. In the case of the p.Pro461Thr protein, K_D_ was 1.61 × 10^−6^ M, and the association rate constant (k_a_) was k_a_ = 1.38 × 10^4^ 1/Ms.

Based on these results, AT–heparin complex formation seems to be much slower in the case of the mutant proteins. Both mutants showed weaker interaction with heparin, and their K_a_ and K_d_ values were similar. The dissociation rate constants (k_d_) in the WT and all mutants were on the same order of magnitude (10^−3^). These observations support the presence of an altered heparin interaction for both mutants.

### 2.8. N-Glycosidase F Digestion of Mutant AT Proteins

As N-linked glycosylation is a crucial post-translational modification involved in protein folding, we used N-glycosylation as a tool to study mutant AT proteins [17]. N-Glycosidase F cleaves asparagine-bound N-glycans from glycoproteins. AT has four N-glycosylation sites (Asn128, Asn167, Asn187 and Asn224); missense variants of these sites and others were reported to inhibit glycosylation [18,19]. N-Glycosidase F digestion was carried out in the cases of AT variants with low amounts of AT in the cell media, suggesting a secretion defect; these variants comprised p.Met121Arg, p.Leu245Pro, p.Asn450Ile and p.Gly456delins_Ala_Thr (Figure 5). Treatment of conditioned media with PNGase altered the electrophoretic migration patterns of the WT and all four variant AT proteins, indicating that these mutant proteins were N-glycosylated in the ER. The WT and the four variant AT proteins had identical electrophoretic migration patterns. Non-digested AT migrated as a single band on SDS-PAGE, with a molecular weight of 58 kDa; PNGase-digested AT was also detected, with a molecular weight of 50 kDa.

## 3. Discussion

In this study, we present the clinical and molecular characterization of nine novel AT mutations (p.Arg14Lys, p.Cys32Tyr, p.Arg78Gly, p.Met121Arg, p.Leu245Pro, p.Leu270Argfs*14, p.Asn450Ile, p.Gly456delins_Ala_Thr and p.Pro461Thr). Indirect and direct pieces of evidence were collected in order to determine their pathogenic nature and to classify them into ATD subgroups. Among missense variants, the p.Met121Arg, p.Leu245Pro, p.Asn450Ile and p.Pro461Thr mutations were predicted as pathogenic by all six in silico methods (PolyPhen2 HumDiv, PolyPhen2 HumVar, MutPred2, PhD-SNP, SIFT and MutationTaster), and the AT sequence homology study also suggested their deleterious effect. Two missense mutations, p.Arg14Lys and p.Cys32Tyr are located in the signal peptide sequence of AT. Two of the in silico methods predicted these mutants to be pathogenic, while two other methods designated them as neutral. The sequence homology study was in accordance with this, since the AT signal peptide sequence was heterogeneous in the seven investigated species. The in silico prediction for the p.Arg78Gly mutant was rather conflicting (three methods predicted it as non-pathogenic), however the homology study indicated a strongly conserved position of this mutation.

The p.Arg14Lys mutation resulted in decreased heparin cofactor anti-FXa AT activity (hc-antiFXa), decreased progressive anti-FXa AT activity (p-antiFXa) and decreased AT antigen concentration in the plasma of the patient, suggesting a rather quantitative ATD. The amount of recombinant p.Arg14Lys AT, however (as shown by results of Western blotting, AT antigen and activity measurements) was not significantly lower than that of the WT AT protein. This and the position of this variant (affecting the N-terminal region of positively charged basic residues in the signal peptide, which is a rather neutral part of it) could not confirm its deleterious nature unequivocally and suggest that the index patient might also have other causative factors in the background of his DVT and also in lower plasma AT levels. Corral and colleagues suggested that in cases without causative *SERPINC1* defects mutations in other genes involved in transcriptional control of the gene or in post-translation modifications or mutations in proteins involved in the clearance of AT might affect the plasma level of AT [1]. It might also happen that mutations in the deep promoter or intronic regions of *SERPINC1*, which are not identified by Sanger sequencing and linked to p.Arg14Lys, are responsible for the lower AT levels. Unfortunately, a large family study could not be executed to further investigate the linkage of this variant to the disease.

The p.Cys32Tyr mutation in the patient resulted in low hc-antiFXa and p-antiFXa AT activity values and a proportionally decreased AT antigen. Western blot analysis of the recombinant p.Cys32Tyr protein detected very low AT protein expression, both in the media and in the cell lysates, and it was confirmed by AT antigen measurements. The RT-qPCR detected a normal mRNA level, suggesting a translational or post-translational defect. The p.Cys32Tyr mutation was located in the conserved polar C-terminal region of the signal peptide containing the cleavage site. In silico analysis of signal peptide predicted an abnormal cleavage site after position 32. The signal peptide plays an important role in translation by ribosomes; the nascent protein is bound via the signal peptide to the signal recognition particle, which guides the complex to the endoplasmic reticulum [20]. Certain positions in the signal peptide are therefore critical from the point of view of signal peptide function. Jochmans et al. described a variant in a patient with type I ATD, where cysteine was replaced with arginine at the same position, and found it pathogenic [21]. We can conclude from our in silico and biochemical studies that p.Cys32Tyr is a deleterious variant leading to type I ATD.

The p.Arg78Gly mutation affects the heparin-binding region of AT. In the case of the patient, the hc-antiFXa AT activity was decreased, while a normal p-antiFXa AT activity and normal AT antigen were detected, suggesting type II HBS AT deficiency. Western blot analysis and AT antigen measurement of the recombinant p.Arg78Gly mutant showed no alteration, however hc-antiFXa was disproportionally decreased. SPR experiments verified an altered heparin–AT interaction. Bravo-Pérez and colleagues identified a variant in the same position, the p.Arg78Gln mutation, which led to an increase in the low-heparin-affinity AT form in plasma. They classified this variant as type II HBS deficiency, which is in accordance to our findings [22].

The p.Met121Arg mutation caused proportionally low hc-antiFXa and p-antiFXa AT activity and AT antigen values in the proband, which suggested type I ATD. Western blot analysis of the recombinant p.Met121Arg protein suggested a secretion disorder, as we detected high AT protein expression in the cell lysate but low in the media. In accordance with this, the AT antigen level was very high in the cell lysate, and it was low in the media compared to the AT level in the cell lysate. We investigated the N-linked glycosylation with N-glycosidase F digestion, it was similar to the WT indicating normal glycosylation in the ER. A hypothetical explanation for the very high AT antigen in the cell lysates could be the unfolded protein response (UPR). Abnormal proteins accumulate in the ER, and the ER stress is buffered by the activation of this mechanism. When the capacity of the UPR to sustain proteostasis is overwhelmed, cells enter the canonical apoptosis pathway [23,24]. In the literature, we found two different mutations in this position, p.Met121Ile [25] and p.Met121Lys [26]; however, no expression studies were carried out in their cases. Our experimental findings support the pathogenic nature of p.Met121Arg mutation.

The p.Leu245Pro mutation was carried by a male patient with proximal DVT. Hc-antiFXa and p-antiFXa AT activity and AT antigen values were proportionally low, suggesting type I ATD. Western blot analysis of the recombinant p.Leu245Pro protein suggested secretion disorder, as we detected high AT protein expression in the cell lysate but low in the media. In accordance with this, the AT antigen level was very high in the cell lysate, and it was low in the media compared to the AT level in the cell lysate. Investigation of the N-linked glycosylation with N-glycosidase F digestion indicated normal glycosylation in the ER. In the literature, we found the AT Murcia (p.Lys241Glu) mutation as a neighboring variant [27]. Certain missense mutations can cause type I deficiencies through their effects on protein folding because misfolded proteins can be degraded in lysosomes or accumulate inside the endoplasmic reticulum [28,29]. Because of the similarity to p.Met121Lys, we hypothesized that the UPR mechanism might cause the type I phenotype.

The p.Leu270Argfs*14 mutation was detected in a male patient with 5 episodes of DVT. From the clinical point of view, this case showed the most severe thrombotic phenotype. Hc-antiFXa and p-antiFXa AT activity and AT antigen values were proportionally low, suggesting type I ATD. Western blot analysis and AT antigen measurement of the recombinant p.Leu270Argfs*14 protein revealed low AT both in the media and in the cell lysates. The RT-qPCR detected a normal RNA level, suggesting a translational defect. After position 270, a stop sequence was generated, causing early termination of translation.

The p.Asn450Ile mutation was carried by a family. Their hc-antiFXa and p-antiFXa AT activity and AT antigen values were proportionally low, suggesting type I ATD. Western blot analysis of the recombinant p.Asn450Ile protein showed low AT protein expression for both the media and the cell lysate. We detected an abnormal AT protein with faster electrophoretic mobility on the SDS-PAGE, which suggests an abnormal post-translational modification [1]. Therefore, we investigated the N-linked glycosylation with N-glycosidase F digestion, but it was similar to the WT, indicating normal N-glycosylation in the ER. Defects in other post-translational mechanisms, however, might be responsible for the development of type I ATD.

The p.Gly456delins_Ala-Thr mutation in the proband, which resulted in a protein that was one amino acid longer, led to proportionally low hc-antiFXa and p-antiFXa AT activity and AT antigen values in accordance with a type I ATD. Western blot analysis of the recombinant p.Gly456delins_Ala-Thr protein showed high AT expression in the cell lysates but low in the media. This result suggested a secretion disorder, but not due to abnormal N-linked glycosylation. Jochmans et al. described a sequence change at position 456, which replaced glycine with arginine (p.Gly456Arg) and was classified as pathogenic [30,31].

The p.Pro461Thr mutation was detected in three unrelated patients. Hc-antiFXa and p-antiFXa AT activity values were disproportionally low compared to AT antigen values, suggesting a rather functional AT deficiency. Western blot analysis and AT antigen measurement of the recombinant p.Pro461Thr protein detected high AT protein levels both in the media and in the cell lysates. Heparin cofactor activity was decreased compared to progressive activity. SPR measurements verified an altered heparin-AT interaction suggesting the presence of an IIPE variant. In accordance with our findings, we found the p.Pro461Leu mutation classified as type IIPE [32,33,34] and the p.Pro461Ser classified as type IIPE in the literature [35].

In the case of novel mutations, confirmation or refutation of their pathogenic nature is important in order to help in patient management. Moreover, by introducing the mutations and their consequences into databases, it will help others with future diagnostics. Because the laboratory assays in ATD have limitations and the AT activity measured in plasma is often not in accordance with the clinical phenotype of the patient it is important to estimate the effect of a novel mutation on the structure and function of AT by in vitro experiments. In our present study, severe type I ATD was confirmed for p.Cys32Tyr, p.Leu270Argfs*14 and p.Asn450Ile (altered synthesis) and for p.Met121Arg, p.Leu245Pro and p.Gly456delins_Ala_Thr (altered secretion). All these patients suffered from at least one episode of thrombosis; moreover, multiple thrombosis was registered in most cases. In case of the type II heparin binding site ATD (p.Arg78Gly), which is suggested to be less severe, the patient did not suffer thrombosis and the diagnosis of ATD was established only upon a throughout laboratory investigation before an in vitro fertilization procedure. She was treated with LMWH during her pregnancy, but no long-term anticoagulation was introduced afterwards, and she is still symptom-free. The p.Pro461Thr mutation has a pleiotropic effect on AT, and its clinical severity is considerable; all patients with this mutation suffered thrombosis. However, either provoking factors were explored in the background, or the patient was older at the time of the thrombotic episode. Finally, the pathogenic role of p.Arg14Lys was equivocal according to our investigations. One of the patients carrying this variant is symptom-free, and the other has suffered one thrombotic episode. It cannot be excluded that a major (not registered) provoking factor was also present in the background of his thrombosis or that the decrease in the AT level was caused by a different, currently unidentified genetic defect in the family.

In summary, in our study, we provided insight into the pathogenic nature of different *SERPINC1* mutations by in vitro expression studies and in silico analysis, by which different mechanisms of pathogenicity were suggested.

## 4. Materials and Methods

### 4.1. Clinical and Routine Laboratory Data of the Antithrombin Deficient Patients

Between January 2016 and December 2021, nine novel AT mutations were detected in our center. Clinical data were collected retrospectively from the patients. Information on the type and date of the first thrombotic event were obtained from medical records. Inherited thrombophilia (protein C and S deficiencies, APC resistance, dysfibrinogenemia) was investigated by routine laboratory methods with a BCS-XP coagulometer (Siemens, Marburg, Germany). For diagnosing AT deficiency hc-anti-FXa and p-anti-FXa (Labexpert Antithrombin H + P, Labexpert Ltd., Debrecen, Hungary; reference intervals 80–120% and 82–118%, respectively) were detected with a Siemens BCS-XP coagulometer. AT antigen was measured by immunonephelometry (Siemens, N Antiserum to Human Antithrombin III, reference interval 0.19–0.31 g/L).

### 4.2. Mutation Analysis of SERPINC1 Gene in Antithrombin Deficient Patients

Genomic DNA was isolated from peripheral whole blood using QIAamp DNA Blood Mini kit (Qiagen GmbH, Hilden, Germany). Sanger sequencing was performed to identify mutations in the exons, the flanking intronic regions and in the promoter of *SERPINC1* gene using an ABI3130 Genetic Analyzer and Sequencing Analysis 5.4 software (Thermo Fisher Scientific, Carlsbad, CA, USA). Multiplex ligation-dependent probe amplification (MLPA) was performed using SALSA MLPA KIT P227 (MRC-Holland, Amsterdam, The Netherlands) using an ABI3130 Genetic Analyzer if Sanger sequencing was negative. The MLPA products were analyzed with GeneMapper Software 4.1 (Thermo Fisher Scientific).

### 4.3. In Silico Prediction of the Consequences of Missense Mutations

To predict whether the seven missense mutations in the present study are disease-causing, we used five different tools available as web servers, resulting in 6 different scores in total. PolyPhen2 is a tool for the prediction of the consequences of amino acid substitutions in human proteins. It takes into account information from multiple sources: homologous sequences, structural features and 3D structures [36]. PolyPhen2 uses two different models trained on different datasets, HumDiv and HumVar. HumVar is better suited for detecting mutations with drastic, disease-causing effect, while HumDiv is intended for predicting more complex phenotypic effects and mildly deleterious variants [37]. MutPred2 uses a machine learning model for classifying mutations in humans as pathogenic or benign. It can also predict the impact of the mutation on several structural and functional properties of the protein [38]. PhD-SNP is a Support Vector Machine (SVM)-based classifier for missense mutations that takes into account the mutation, the sequence environment and the sequence profile [39].

SIFT is a sequence-homology-based method for distinguishing between amino acid substitutions in proteins that with our without a phenotypic consequence. Mutations with a score below or equal to 0.05 are classified as damaging [40]. MutationTaster is a tool that predicts the consequences of variants and mutations in DNA sequences. In MutationTaster, the score is the probability of correct prediction and not a value that distinguishes between pathogenic and non-pathogenic mutants [41]. For the PolyPhen2, MutPred2 and PhD-SNP methods, mutations with a score above a cut-off of 0.5 were considered pathogenic. For the two missense mutants located in the signal peptide, p.Arg14Lys and p.Cys32Tyr, we also performed signal peptide cleavage prediction using SignalP 6.0, for eukaryotic signal peptide only [42].

### 4.4. In Vitro Expression of Wild Type and Mutant Antithrombins

The cDNA clone ORF-NM_000488_pcDNA3.1(+) wild-type AT (WT) was purchased from ImaGenes GmbH (Berlin, Germany). The nine mutant plasmids (p.Arg14Lys, p.Cys32Tyr, p.Arg78Gly, p.Met121Arg, p.Leu245Pro, p.Leu270Argfs*14, p.Asn450Ile, p.Gly456delins_Ala_Thr and p.Pro461Thr) were created by us using the QuickChange Site-Directed Mutagenesis (Agilent Technologies, Santa Clara, CA, USA) kit according to the manufacturer’s instructions. HEK293 cells were grown in Dulbecco’s Modified Eagle’s Medium (DMEM, High glucose, Biosera) supplemented with 10% fetal bovine serum (FBS, Gibco), 2 mM L-glutamine and 25 μg/mL gentamicin antibiotic (Chinoin, Budapest, Hungary) at 37 °C and 5% CO_2_ in a humidified incubator. For all experiments, cells were grown to 60–80% confluency and were subjected to no more than 20 cell passages. Cells were subcultured every 3 days using a standard trypsinization procedure. Transient transfection of the WT and the nine mutant AT plasmids was performed using X-tremeGENE HP DNA Transfection Reagent (Roche Diagnostics GmbH, Mannheim, Germany) and co-transfection of the LacZ gene was also performed with pCMV Sport β-GAL plasmid (Invitrogen, Carlsbad, CA, USA). After 48 h of incubation, conditioned media were collected, and the cells were lysed in a buffer containing 50 mM Tris-HCl (pH 7.5), 150 mM NaCl, 1% Nonidet P40, 0.5% sodium deoxycholate and a protease inhibitor cocktail (Roche). A FluoReporterlacZ/Galactosidase Quantitation Kit (Molecular Probes, Life Technologies) was used to investigate the transfection efficiency, and the results were corrected accordingly.

### 4.5. Stable Transfection of Wild-Type and Mutant Antithrombins

Stable transfection of HEK-293 cells with huSERPINC1_pcDNA3.1(+) wild_type and huSERPINC1_pcDNA3.1(+)_p.Arg14Lys _type, _ p.Cys32Tyr_type, _ p.Arg78Gly_type, _ p.Met121Arg_type, _ p.Leu245Pro_type and _ p.Pro461Thr_type plasmids, respectively, was performed as follows. The first step was the generation of a kill curve to determine the optimal selection antibiotic concentration for selecting stable cell colonies. A kill curve is a dose–response experiment where the HEK293 cells are subjected to increasing amounts of geneticin (0–1000 µg/mL) to determine the minimum concentration that kills all the cells within 10 days. We used Geneticin^®^ Selective Antibiotic (Gibco, Thermo Fisher Scientific, Carlsbad, CA, USA) as a selective agent in our stable transfection experiments. Resistance to geneticin is conferred by the neomycin resistance gene which is in the pcDNA3.1(+). We found a 400 µg/mL geneticin concentration optimal. The second step was the transfection of the wild type and the six mutant antithrombin plasmids. Cells were plated in a 24-well dish (reaching about 70–80% confluence) and transfected the following day. Transfection was performed by addition of 2.0 μg plasmid DNA encoding for the wild-type and the mutant AT in reduced serum Opti-MEM medium (Gibco, Thermo Fisher Scientific, Carlsbad, CA, USA) with a Lipofectamine^®^ 3000 Transfection Kit (Invitrogen, Carlsbad, CA, USA) according to the manufacturer’s instructions. The third step was the selection for transfected cells. After 24 h, 400 µg/mL geneticin was added to select the resistant colonies. The selection medium was replaced every three days. We cultured the cells in selective medium for ten days. Most of the cells that had not integrated the transfected plasmid died, while the cells that had undergone plasmid integration survived. The surviving cells were allowed to expand; when the cells in the T75 culture flask reached high confluence, we froze them as a polyclonal line.

### 4.6. AT Antigen and Activity Measurements in the Conditioned Media Containing Each Recombinant AT Protein

Aliquots of the conditioned media and cell lysates were used for AT antigen determination by ELISA (Abcam Human-Antithrombin-III-ELISA-Kit, Abcam, Cambridge, UK). AT activity from the conditioned media of the transfected cells was detected through an amidolytic assay in a microtiter plate using LX Antithrombin Hc + P, FXa reagent (Labexpert Ltd., Debrecen, Hungary) with minor modifications. Briefly, this assay comprised bovine FXa as a substrate and BIOPHEN CS-11(32) [Suc-Ile-Gly-(γ Pip)Gly-Arg-pNA, HCl] as a chromogenic substrate (HYPHEN Biomed., Neuville, France). We used a final dilution of 1:2 of conditioned media of transfected HEK293 cells in a Tris-HCl buffer (pH 8.4) containing heparin (heparin cofactor activity). We performed the assay with the same conditions in the absence of heparin (progressive activity).

### 4.7. N-Glycosidase F Digestion

N-Glycosidase F cleaves asparagine-bound N-glycans from glycoproteins. The AT variants (p.Met121Arg, p.Leu245Pro, p.Asn450Ile and p.Gly456delins_Ala_Thr) were treated with N-Glycosidase F (Roche Diagnostics GmbH, Mannheim, Germany). Before PNGase F digestion, the samples were pretreated with 150 mM Na-phosphate at 95 °C for 5 min. Then, we added 6 μL 100 U/mL PNGase F enzyme and incubated the samples at 37 °C for 15 h. Samples were resolved on 10% SDS-PAGE gels and then transferred to nitrocellulose membranes. Through Western blot analysis, the AT specific immunoreactive bands were visualized with enhanced chemiluminescence (ECL) detection (Thermo Fisher Scientific, Carlsbad, CA, USA).

### 4.8. Western Blot Analysis

Conditioned media were collected; then, the cells were lysed in a buffer containing 50 mM Tris-HCl (pH 7.5), 150 mM NaCl, 1% Nonidet P40, 0.5% sodium deoxycholate and a protease inhibitor cocktail (Roche) and centrifuged at 12,000× *g* for 15 min at 4 °C. The total protein concentrations of the extracts were measured with a Pierce BCA protein assay kit (Thermo Fisher Scientific). Equal amounts of protein were loaded onto 10% SDS-PAGE gels and then transferred to nitrocellulose membranes. After the membranes were blocked at room temperature for 1 h in 5% nonfat dry milk in TBS-T buffer, they were incubated overnight with primary antibodies at 4 °C. AT was detected with a goat anti-human AT antibody (Affinity Biologicals, Ancaster, ON, Canada); beta tubulin polyclonal antibody (Invitrogen) was used as a loading control. The primary antibodies were diluted 1:10,000 in blocking solution. The membranes were washed three times for 7 min with TBS-T buffer and incubated for 45 min at room temperature with the appropriate secondary antibody. Horseradish peroxidase-conjugated anti-goat IgG (Abcam) and anti-rabbit IgG (GE Healthcare) were diluted 1:10,000 in blocking solution. Immunoreactive bands were visualized with ECL following the manufacturer’s instructions (Thermo Fisher Scientific). Chemiluminescent imaging was performed with a C300 Azure Imaging system (Azure Biosystems, Dublin, CA, USA).

### 4.9. Real-Time Quantitative PCR Analysis

We performed real-time quantitative RT-PCR to investigate the mRNA expression levels of the wild-type form and the nine mutant AT forms in transient transfection. Total RNA was isolated from transiently transfected HEK293 cells with QIAamp RNA Blood Mini Kit (QIAGEN). We used a RapidOut DNA Removal Kit (Thermo Scientific) to remove the genomic DNA, and after reverse transcription (qPCRBIO cDNA Synthesis Kit), real-time quantitative PCR was performed using the following primers: AT RNA forward primer 5′ –GCTAAACCCCAACAGGGTGA-3′; AT RNA reverse primer 5′ –TTACTTAACACAAGGGTTGGCTAC-3′; OAZ RNA forward primer 5′ –CACCATGCCGCTCCTAAG-3′; OAZ RNA reverse primer 5′ –GAGGGAGACCCTGGAACTCT-3′; β-GAL RNA forward primer 5′ –GCGTACATCGGGCAAATAAT-3′; β-GAL RNA reverse primer 5′ –TAATCACGACGCGCTGTATC-3′.

### 4.10. Preparation of Antithrombin from In Vitro-Expressed Recombinant Antithrombins by Affinity Chromatography

Media containing expressed WT and mutant AT (p.Arg78Gly and p.Pro461Thr) proteins were harvested and concentrated on an Amicon^®^ Ultra 30 kDa (Merck Millipore, Burlington, VT, USA) column. Then, AT proteins were purified by affinity chromatography using goat anti-human antithrombin IgG (Affinity Biologicals, Ancaster, ON, Canada) that was covalently coupled to Sepharose 4B gel.

### 4.11. Surface Plasmon Resonance

Surface plasmon resonance (SPR) assays were performed on a Biacore 3000 instrument (GE Healthcare, Uppsala, Sweden). We used a heparin-coated SPR sensor chip (Heparin Approx. 50 nm hydrogel chip, XanTec bioanalytics GmbH, Dusseldorf, Germany) for assaying the binding characteristics between different AT mutants and heparin. Wild-type and mutant AT were diluted in running buffer (HEPES 10 mM, NaCl 150 mM, EDTA 3 mM, surfactant 0.005% (*v*/*v*), pH 8.4) and injected over the heparin-coated sensor chip surface at 6 different concentrations (40, 80, 120, 160, 200 and 240 nM) at a flow rate of 10 μL/min for 7 min. Between two measurements, sensor chip surfaces were regenerated with 30 μL regeneration buffer (10 mM glycine-HCl, pH 2.5, GE Healthcare, Uppsala, Sweden). The optimal pH of the interaction was previously determined; thus, the analysis proceeded at pH 8.4 [43]. A Langmuir 1:1 binding model was used for curve fitting. The rate constants for association and dissociation (ka and kd) as well as the equilibrium constants for association and dissociation (KA and KD) were calculated from the sensorgrams. BIAevaluation software version 3.2 (GE Healthcare, Uppsala, Sweden) was used to evaluate the interaction analysis.

## Figures and Tables

**Figure 1 ijms-25-02893-f001:**
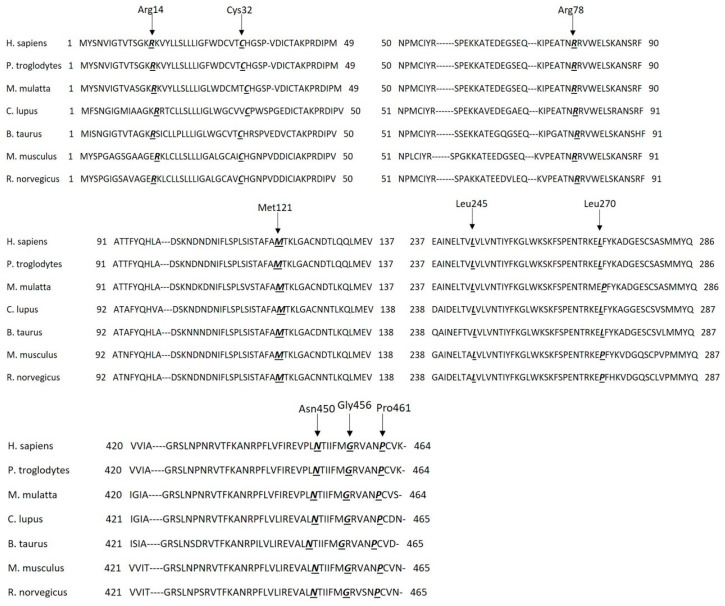
AT sequence homology in seven different species according to UniProt Database (http://www.uniprot.org). Single-letter code for amino acids: Ala, A; Arg, R; Asn, N; Asp, D; Cys, C; Gln, Q; Glu, E; Gly, G; His, H; Ile, I; Leu, L; Lys, K; Met, M; Phe, F; Pro, P; Ser, S; Thr, T; Trp, W; Tyr, Y; Val, V. Species: *H. sapiens*, *Homo sapiens*; *P. troglodytes*, *Pan troglodytes*; *M. mulatta*, *Macaca mulatta*; *C. lupus*, *Canis lupus*; *B. taurus*, *Bos taurus*; *M. musculus*, *Mus musculus*; *R. norvegicus*, *Rattus norvegicus*. The numbers are residue numbers in the sequence of each protein. Amino acid residues corresponding to the mutations in our patients are labeled in bold italics, and they are underlined.

**Figure 2 ijms-25-02893-f002:**
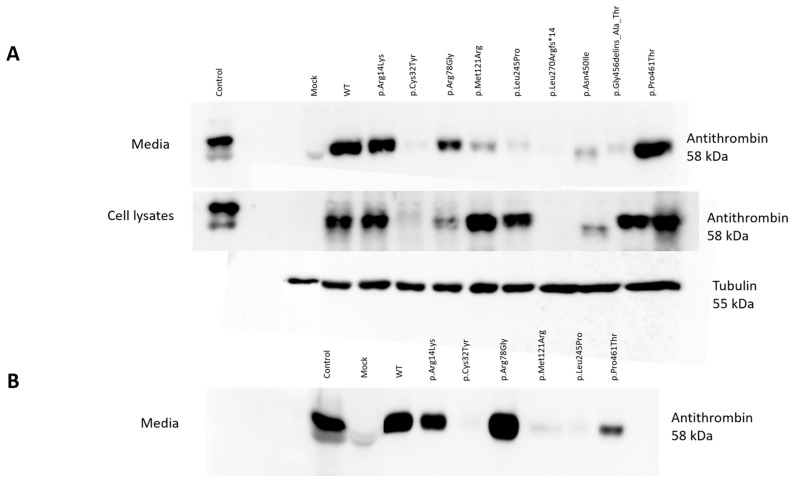
Detection of wild-type and mutant antithrombins by Western blotting in the media and lysates of transiently (**A**) and stably (**B**) transfected HEK293 cells. SDS-PAGE was performed in non-reducing conditions. In the case of cell lysates, tubulin (55 kDa) served as an internal control. Mock-transfected cells served as a negative control, and 40-fold diluted pooled plasma was used as a positive control.

**Figure 3 ijms-25-02893-f003:**
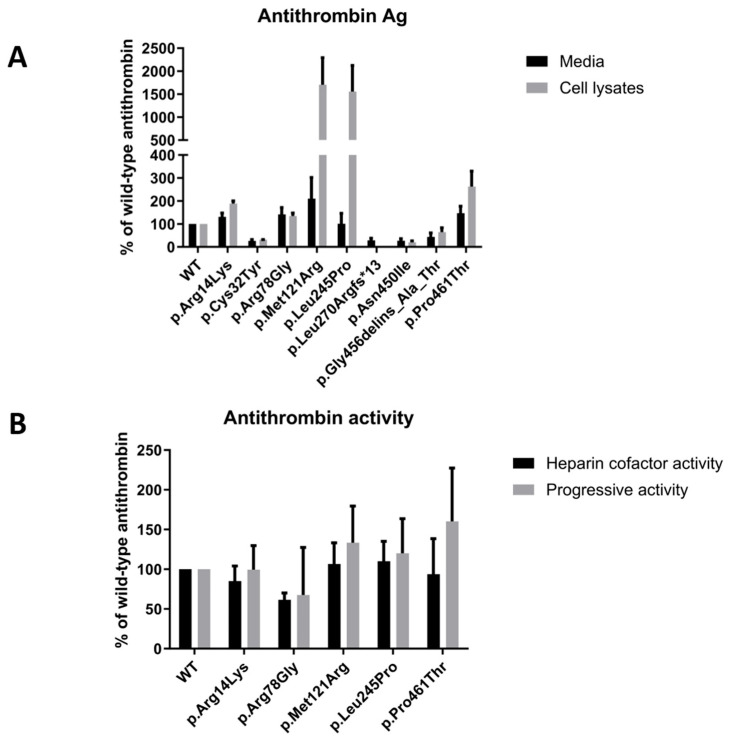
AT antigen and activity of the different mutants expressed in HEK293 cells. (**A**) AT antigen measurement in media and cell lysates of transfected HEK293 cells according to ELISA. Results were normalized for transfection efficiency (transient transfection) and expressed as a percentage of the WT AT antigen. In the graph, we show the average of the measurements obtained from different transfections; error bars represent the standard error of the mean. Black columns indicate AT concentration in the media, and gray columns indicate AT concentration in the cell lysates. (**B**) AT activity measurements (heparin cofactor activity, hc-antiFXa; marked as the black column; progressive activity, p-antiFXa, marked as the gray column) were determined in the conditioned media of transiently transfected HEK293 cells through an amidolytic assay.

**Figure 4 ijms-25-02893-f004:**
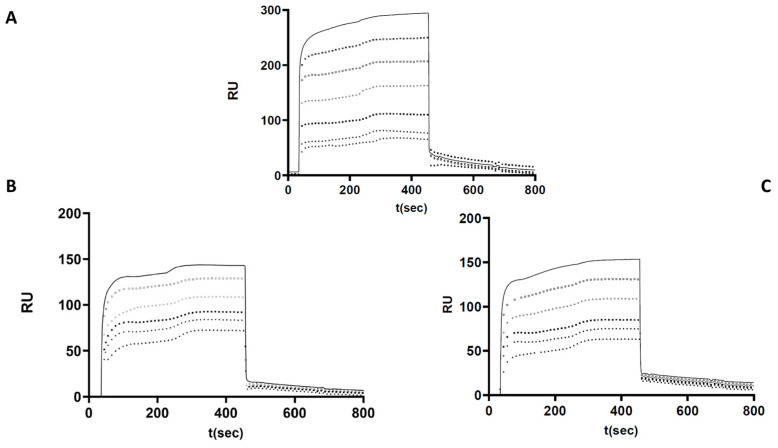
Interaction analysis of wild-type and mutant antithrombin proteins with heparin-coated sensor chip surfaces using a surface plasmon resonance-based binding technique. Wild-type and mutant AT proteins were purified from the media of transfected HEK293 cells by affinity chromatography. AT forms at different concentrations (40, 80, 120, 160, 200 and 240 nM from bottom to top curves in the pictures) were injected over the heparin-coated sensor chip surface. Sensorgrams corresponding to the wild-type form and two different mutant AT forms are shown in panels (**A**–**C**): (**A**) wild-type AT, (**B**) p.Arg78Gly AT and (**C**) p.Pro461Thr AT.

**Figure 5 ijms-25-02893-f005:**
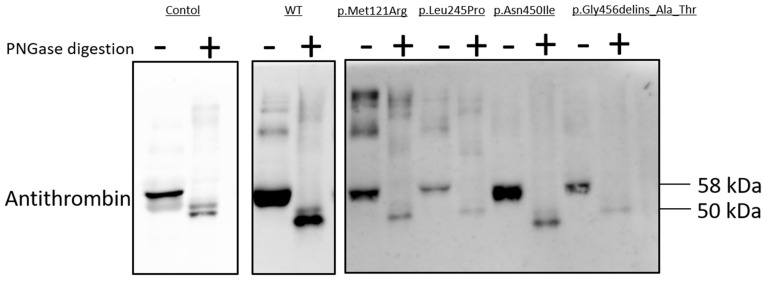
N-glycosidase F digestion of mutant AT proteins. Western blot analysis of the conditioned media after PNGase digestion. SDS-PAGE was performed in non-reducing conditions, and 40-fold diluted pooled plasma was used as a positive control.

**Table 1 ijms-25-02893-t001:** Genotype–phenotype correlations in patients.

No.	Nucleotide Position (Nomenclature According to HGVS)	Amino Acid Position (Nomenclature According to HGVS)	Patient ID	Gender	hc-antiFXa Activity (%)	p-antiFXa Activity (%)	AT Antigen (g/L)/ (%)	Symptoms	Age at First Onset (Year)	Age at Diagnosis (Year)	Provoked Thrombosis (Yes/No)	Recurrent Thrombotic Events (Yes/No)	Anticoagulant Therapy
1	c.41G > A	p.Arg14Lys	1_1	M	63	79	0.16/64	1 DVT	16	23	No	No	VKA
			1_2 (father of 1_1)	M	53	76	0.13/52	No	-	47	-	-	No
2	c.95G > A	p.Cys32Tyr	2_1	F	60	75	0.15/60	2 DVT, 1 PE	20	41	Yes, pregnancy	Yes	RIVA 1 × 20 mg
3	c.232C > G	p.Arg78Gly	3_1	F	54	94	0.27/108	No	-	34	-	-	No
4	c.362T > G	p.Met121Arg	4_1	F	52	53	0.13/52	1 DVT	34	41	No	No	VKA
5	c.734T > C	p.Leu245Pro	5_1	M	59	65	0.18/72	1 DVT	42	43	No	No	VKA
6	c.809delT	p.Leu270Argfs*14	6_1	M	56	60	0.16/64	5 DVT	16	59	No	Yes	RIVA 1 × 20 mg
7	c.1349A > T	p.Asn450Ile	7_1	F	49	52	0.14/56	No	-	27	-	-	No
			7_2 (sister of 7_1)	F	55	60	0.15/60	1 DVT	23	30	no	No	VKA
			7_3 (father of 7_1)	M	59	62	0.16/64	2 DVT	40	55	no	Yes	VKA
			7_4 (cousin of 7_1)	M	60	59	0.15/60	No	-	27	-	-	No
8	c.1367-1368delGCinsCTACA	p.Gly456delins_Ala-Thr	8_1	F	54	46	0.15/60	2 DVT	27	33	Yes, pregnancy	Yes	API 2 × 5 mg
9	c.1381C > A	p.Pro461Thr	9_1	F	31	ND	0.21/84	1 DVT + PE	62	62	No	No	RIVA 1 × 20 mg
			9_2	M	55	58	0.18/72	1 DVT	48	64	Yes, long flight travel	No	VKA
			9_3	F	55	62	0.18/72	1 DVT	27	30	Yes, OAC	No	API 2 × 5 mg

Abbreviations: HGVS, Human Genome Variation Society; ND, no data; F, female; M, male; DVT, deep vein thrombosis; PE, pulmonary embolism; OAC, oral contraceptives; VKA, vitamin K antagonist; API, apixaban; RIVA, rivaroxaban. Reference intervals for hc-antiFXa activity, p-antiFXa activity and AT antigen are 80–120%, 82–118% and 0.19–0.31 g/L, respectively. Patients No 2_1 and No 8_1 are heterozygous for FV Leiden. No other thrombophilic conditions were registered in our patients.

**Table 2 ijms-25-02893-t002:** In silico prediction of the consequences of missense mutations.

	PolyPhen2 HumDiv	PolyPhen2 HumVar	MutPred2	PhD-SNP	SIFT	MutationTaster
p.Arg14Lys	Probably damaging (0.994)	Probably damaging (0.970)	Non-pathogenic (0.273)	Neutral (0.355)	Affects protein function * (0.00)	Polymorphism (0.7298)
p.Cys32Tyr	Probably damaging (0.981)	Possibly damaging (0.635)	Non-pathogenic (0.415)	Disease (0.868)	Affects protein function * (0.00)	Polymorphism (0.6706)
p.Arg78Gly	Benign (0.126)	Benign (0.031)	Pathogenic (0.665)	Neutral (0.371)	Affects protein function * (0.00)	Disease-causing (1.0000)
p.Met121Arg	Probably damaging (1.000)	Probably damaging (1.000)	Pathogenic (0.944)	Disease (0.889)	Affects protein function (0.00)	Disease-causing (1.0000)
p.Leu245Pro	Probably damaging (1.000)	Probably damaging (1.000)	Pathogenic (0.952)	Disease (0.897)	Affects protein function (0.00)	Disease-causing (1.0000)
p.Asn450Ile	Probably damaging (1.000)	Probably damaging (0.999)	Pathogenic (0.839)	Disease (0.840)	Affects protein function (probability: 0.01)	Disease-causing (1.0000)
p.Pro461Thr	Probably damaging (0.999)	Probably damaging (0.972)	Pathogenic (0.724)	Disease (0.703)	Affects protein function (probability: 0.00)	Disease-causing (1.0000)

* SIFT reported lower reliability for these predictions.

## Data Availability

The datasets generated for this study are available on request from the corresponding author.

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
