# Peer review of "Clinical and Molecular Characterization of Nine Novel Antithrombin Mutations"

_ijms, 2024, doi:10.3390/ijms25052893_

Round 1
Reviewer 1 Report
Comments and Suggestions for Authors
This is an exciting research paper.
However, a few suggestions are placed that need to be clarified or included to further improve the manuscript.
Comment 1: Introduction is too long, it should be concise.
Comment 2: comparison with healthy controls would have given better insight ? If yes, kindly add in limitations.
Comment 3: was there any other prothrombotic risk factors in these patients ?
Comment 4: what is the prevalence of AT mutations in present study? Is it higher than the published literature?
Comment 5: Towards the end, please mention, what will be the future use of these data or how will these findings be clinically relevant?
Comments on the Quality of English Language
Looks good!
Author Response
We are thankful for the reviewer for the comments and questions. We have modified the manuscript according to the requirements.
Q1. Introduction is too long, it should be concise.
Answer: Thank you for the comment, we have shortened the introduction.
Q2. comparison with healthy controls would have given better insight ? If yes, kindly add in limitations.
Answer: Since antithrombin deficiency is a rare disease it is difficult to conduct an epidemiology study with efficient statistical power. In our case-based study on the effects of novel mutations we only had a chance to describe the patients’ characteristics and to perform in vitro study to investigate the biochemical effects of the nine SERPINC1 mutations on antithrombin. In these in vitro studies we used wild type antithrombin expressed in HEK293 cells for comparison. However we extended the clinical description of the patients with additional health condition data and data on anticoagulation.
Q3. was there any other prothrombotic risk factors in these patients ?
Answer: Thank you for the question. We have performed complex thrombophilia screening in all patients and no additional thrombophilia was found except for two female patients (No 2_1 and 8_1 in Table 1) who were heterozygous for FV Leiden mutation. This is indicated now in the revised version of the manuscript in the legend of Table 1. Antiphospholipid syndrome was not found in any of the patients. Concerning provoking factors there were two patients with pregnancy-related thrombosis and one female with oral contraceptive pills, and one patient suffered thrombosis after a long flight travel, which are indicated in Table 1. Other risk factors were not registered.
Q4. what is the prevalence of AT mutations in present study? Is it higher than the published literature?
Answer: This is an interesting question, however clear answer can not be given for this for several reasons. First of all, our Cinical Center serves as a regional center for hemostasis disorders including rare diseases, therefore patients are received from a relatively large area, although the whole country is not covered. From the point of view of genetic investigations our center covers the whole country, however genetic tests are not requested in many patients with antithrombin deficiency, which cases are lost from the genotype-based study, like this. As being a regional center collecting AT deficient cases the frequency of AT mutations seems higher for us. However, taking into consideration that the population of Hungary is currently 9.8 million (demographic data of Hungary 2023) the average prevalence of antithrombin deficiency is suggested not being higher than the published literature. It is to be noted, that in most antithrombin deficient cases the founder AT Budapest 3 mutation is found in our population having many cases with extremely severe homozygous conditions. The AT Budapest 3 families represent the majority of antithrombin deficiency especially in the eastern region of the country (in the surrounding area of our Center), where the prevalence of antithrombin deficency is definitely higher than in other regions of the country. Due to the lack of antithrombin deficiency national registry correct prevalence data unfortunately are not provided.
Q5. Towards the end, please mention, what will be the future use of these data or how will these findings be clinically relevant?
Answer: In case of novel mutations it is important to confirm or rebut their pathogenic nature and so they can be introduced into the genetic databases helping others for future diagnostics. Since the laboratory assays in antithrombin deficiency have limitations, and the antithrombin activity measured in plasma is often not in accordance with the clinical phenotype of the patient it is important to estimate the effect of a novel mutation on the structure-function of antithrombin by in vitro experiments. Once the mutation has a deleterious effect on antithrombin, then a severe deficency is to be considered and long-term anticoagulant prophylaxis is suggested. If the pathogenic nature of a mutation is equivocal or not confirmed then long-term anti-thrombotic treatment may not be considered. As thrombosis is a multifactorial disorder the individual risk is dependent on the presence of several factors. If a very severe thrombophilia is demonstrated in a patient the presence of additional risk factors becomes less important from the point of view of thrombosis risk. In our present study severe type I ATD was confirmed for p.Cys32Tyr, p.Leu270Argfs*14, p.Asn450Ile (altered synthesis) and for p.Met121Arg, p.Leu245Pro, and p.Gly456delins_Ala_Thr (altered secretion). All these patients suffered from at least one episode of thrombosis, moreover multiple thrombosis was registered in most cases. In case of the type II heparin binding site ATD (p.Arg78Gly), which is suggested to be less severe, the patient did not suffer thrombosis and the diagnosis of ATD was established only upon a throughout laboratory investigation before in vitro fertilization procedure. She was treated with LMWH during her pregnancy, but no long-term anticoagulation was introduced afterwards and she is still symptom-free. The p.Pro461Thr has a pleiotropic effect on antithrombin and the clinical severity of it is considerable, all patients with this mutation suffered thrombosis, however either provoking factors were explored in the background or the patient was older at the time of the thrombotic episode. Finally, the pathogenic role of p.Arg14Lys was equivocal according to our investigations. One of the patients carrying this variant is symptom-free, and the other has been suffered one thrombotic episode. It cannot be excluded that in the background of his thrombosis a major (not registered) provoking factor was also present, or the decrease in AT level was caused by an other – not yet identified -genetic defect. These considerations are now added to the discussion of the manuscript.
We hope that the modified version of our manuscript is now acceptable for publication.
Reviewer 2 Report
Comments and Suggestions for Authors
Manuscript ID: ijms-2890149
Title: Clinical and Molecular Characterization of Nine Novel Antithrombin Mutations
Authors: Judit Kállai et al.
The manuscript is written in good English. Introduction presents in depth the current state of knowledge about antithrombin and its deficiencies. Similarly, the clinical and experimental part of the manuscript concerning the authors’ own research does not raise any objections. There are only minor inaccuracies such as:
- Page 2 , line 54. The numbering of references presented together should be in numerical order. If reference 4 refers to the first part of the sentence and reference 2 to the refers to the second part, the references numbers should be separated. Reference 4 should be inserted into the text after the part of the sentence relating to it. And then reference 2 after the part of text supported by this reference.
- Page 2, lines 58-59. The text should be unambiguous, therefore “failure to maintain” should be replaced by “a decrease in”.
The above comments lead to the suggestion that it would be advisable to re-read the entire text of the manuscript by all co-authors before its publication.
Author Response
We are thankful for the reviewer for the comments and suggestions. We have modified the manuscript according to the requirements.
Q1.
- Page 2 , line 54. The numbering of references presented together should be in numerical order. If reference 4 refers to the first part of the sentence and reference 2 to the refers to the second part, the references numbers should be separated. Reference 4 should be inserted into the text after the part of the sentence relating to it. And then reference 2 after the part of text supported by this reference.
Answer: Thank you for the suggestion, we have made the corrections.
Q2.
- Page 2, lines 58-59. The text should be unambiguous, therefore “failure to maintain” should be replaced by “a decrease in”.
Answer: Thank you for this correction, we have replaced the text accordingly.
Reviewer 3 Report
Comments and Suggestions for Authors
Content suggestions:
1. For the completness, I would like to kindly ask the Authors whether they can provide the results of further coagulation tests (either standard – aPTT, INR, Fbg level... and also specific (D-dimers...).
2. I am aware of the fact that the manuscript was written predominantly with the aim to report on the new nine mutations of gene encoding antithrombin synthesis. However, can the Authors provide details about the treatment of included patients and the contribution of the knowledge about new mutations for the management of such patients – I suppose that potential further VTE event would be treated by DOACs or VKAs...
The text is written in standard English language and on a high scientific level. I sincerely appreciate each new information about the antithrombin deficiency, as it is one of the most severe thrombophilic states endangering the patient´s life by the development of life-threatening venous thromboembolism.
Please, do not evaluate my comments badly. I was just wondering if you can provide more data about the tested patients.
From my point of view, after the implementation of the responses to the questions of the reviewers and its minor revision, the manuscript can be accepted for publication.
Author Response
We are thankful for the reviewer for the comments and suggestions. We have modified the manuscript according to the requirements.
- For the completness, I would like to kindly ask the Authors whether they can provide the results of further coagulation tests (either standard – aPTT, INR, Fbg level... and also specific (D-dimers...).
Answer: Thank you for the question. As the basic laboratory tests (eg. aPTT, INR, Fbg) for the different patients were performed in different hospitals the results of these tests are not comparable. Moreover – from the point of view of the screening test of coagulation – the results might be affected by the anticoagulant drug administered to the patient at the time of blood collection. Therefore we can just mention (based on the patients’ records) that all basic tests of coagulation were normal or at least appropriate for the anticoagulant therapy of the patients (the anticoagulant therapy is now indicated in Table 1 in the revised version). We added this information about basic tests to the text. Concerning specific tests D-dimer were also measured in different hospitals and according to the patients’ records no elevated D-dimer was described in any of them after the acute phase of thrombosis and under anticoagulant treatment. We investigate the patients for thrombophilia and except for two patients no additional thrombophilia was present. We found APC resistance, FV Leiden heterozygous state in two ladies (patients No 2_1 and 8_1 in Table 1). We also added a text with this information to the manuscript.
- I am aware of the fact that the manuscript was written predominantly with the aim to report on the new nine mutations of gene encoding antithrombin synthesis. However, can the Authors provide details about the treatment of included patients and the contribution of the knowledge about new mutations for the management of such patients – I suppose that potential further VTE event would be treated by DOACs or VKAs...
Answer: Thank you for this question. We added the treatment of the patients to Table 1 and also made a comment on this issue in the text. The patients, who already suffered from at least one episode of thrombosis have been put on long-term anticoagulation by VKA or DOACs. ATD patients during pregnancy (patients No 3_1, 7_1 and 7_2 in Table 1) were treated by LMWH with AT concentrate supplementation. After delivery patient 7_2 was put on VKA for long-term, while pts 3_1 and 7_1 stopped anticoagulation later. We added a detailed description of anticoagulation of our ATD patients to the text.
The text is written in standard English language and on a high scientific level. I sincerely appreciate each new information about the antithrombin deficiency, as it is one of the most severe thrombophilic states endangering the patient´s life by the development of life-threatening venous thromboembolism.
Please, do not evaluate my comments badly. I was just wondering if you can provide more data about the tested patients.
From my point of view, after the implementation of the responses to the questions of the reviewers and its minor revision, the manuscript can be accepted for publication.
Answer: Thank you for your comments. We tried to answer all questions according to our data registered for the patients and we added more detailed description of the cases (please see in the text). We hope that with these modifications our paper is accepted for publication.